# Survival Outcomes of Immune Checkpoint Inhibitors in Conjunction with Cranial Radiation for Older Adults with Non-Small Cell Lung Cancer and Synchronous Brain Metastasis

**DOI:** 10.3390/curroncol32090499

**Published:** 2025-09-05

**Authors:** Ruchira V. Mahashabde, Sajjad A. Bhatti, Bradley C. Martin, Jacob T. Painter, Mausam Patel, Analiz Rodriguez, Jun Ying, Chenghui Li

**Affiliations:** 1Division of Pharmaceutical Evaluation and Policy, College of Pharmacy, University of Arkansas for Medical Sciences, Little Rock, AR 72205, USA; rvmahashabde@uams.edu (R.V.M.); bmartin@uams.edu (B.C.M.); jtpainter@uams.edu (J.T.P.); 2Division of Hematology and Medical Oncology, Department of Internal Medicine, College of Medicine, University of Arkansas for Medical Sciences, Little Rock, AR 72205, USA; sajjad.bhatti@sluhn.org; 3Department of Radiation Oncology, University of Arkansas for Medical Sciences, Little Rock, AR 72205, USA; mpatel2@uams.edu; 4Department of Neurosurgery, University of Arkansas for Medical Sciences, Little Rock, AR 72205, USA; arodriguez@uams.edu; 5Department of Biostatistics, University of Arkansas for Medical Sciences, Little Rock, AR 72205, USA; jun.ying@cuanschutz.edu; 6Department of Family Medicine, School of Medicine, University of Colorado, Aurora, CO 80045, USA

**Keywords:** immune checkpoint inhibitors, cranial radiation, non-small cell lung cancer, brain metastasis, older adults, concurrent treatment

## Abstract

Brain metastases are a serious complication of non-small cell lung cancer (NSCLC) that often lead to poor survival. Combining immunotherapy with brain radiation could be more effective than either treatment alone, but the optimal timing of this combination is unknown. Using real-world data from older adults who had brain metastases at diagnosis of NSCLC, we compared surival outcomes of the combination therapy that was administered within different timing windows. This population is largely underrepresented in clinical trials. Our findings could help clinicians optimize combination treatment strategies for these patients who are at high risk of poor survival outcomes.

## 1. Introduction

Lung cancer remains the leading cause of cancer-related death in the United States [1]. There are two main types of lung cancer: small cell lung cancer, which accounts for 10–15% of patients, and non-small cell lung cancer (NSCLC), which accounts for 80–85% [2]. Unlike small cell lung cancer, which exhibits neuroendocrine features, NSCLC is histologically and molecularly heterogeneous, and it includes adenocarcinoma squamous cell carcinoma and other subtypes [3]. A critical challenge for treating NSCLC is brain metastases (BMs), which affect up to 40% of patients and confer a predicted survival of only 3–15 months after diagnosis [4,5]. Approximately 10% of patients with NSCLC have “synchronous BM” (SBM) (i.e., BM at diagnosis) [6], which accounts for up to 80% of all SBM cases [5]. Not only is lung cancer the most common SBM-associated cancer, but it also has the worst prognosis, with a median survival of approximately 3.9 months and only 20% survival rate beyond the first year [5].

While SCLC initially responds better to chemotherapy, NSCLC’s genomic diversity has allowed targeted therapies and immune checkpoint inhibitors (ICIs) to transform treatment of advanced NSCLC [7]. ICIs enhance immune-mediated destruction of cancer cells via T-cell activation [8], improving overall survival (OS) for patients with locally advanced or metastatic NSCLC, either as monotherapy [9,10,11,12] or in combination with chemotherapy [13,14,15]. BM may benefit from ICI treatment because half of the NSCLC BMs were found to display PD-L1 expression [16]. Despite this biomarker association and high prevalence of BMs in solid tumors, clinical trials often exclude patients with active or untreated BM. Ad hoc analyses of trials that included stable or treated BMs reported an OS benefit among these patients [13,17,18], further supporting potential efficacy of ICIs in the central nervous system, which historically was considered to be immune privileged [19,20]. Cranial radiation (CR) is standard care for BM; stereotactic radiosurgery (SRS) is recommended for limited BMs, and whole-brain radiation therapy (WBRT) with neuroprotection is recommended for extensive disease [21,22]. Preclinical studies have shown that radiation upregulates immune checkpoints, such as PD-1 on tumor-infiltrating T cells and PD-L1 on cancer cells [23], peaking approximately 3 days after radiation treatment [24]. These immunomodulatory properties of CR, combined with immune-enhancing effects of ICIs, suggest a promising synergistic potential, as previously reported in preclinical and clinical studies [25,26].

However, clinical trial data on ICI–radiation synergy in NSCLC are limited to extracranial radiotherapy and with mixed findings. The PACIFIC trial showed improved progression-free survival and OS among patients with unresectable stage 3 NSCLC who received durvalumab within 42 days after completion of chemoradiation [27]. A phase 2 trial showed enhanced event-free 4-year survival among patients with early-stage NSCLC who were undergoing stereotactic ablative radiotherapy with nivolumab, compared to stereotactic ablative radiotherapy alone [28]. However, several recent phase III trials that evaluated the addition of ICI to radiation therapy in early and locally advanced NSCLC failed to demonstrate a survival benefit. The S1914 trial [29], which tested neoadjuvant concurrent and adjuvant atezolizumab use, with stereotactic body radiation therapy (SBRT) versus only SBRT in stage I/II NSCLC, showed no survival benefit but more adverse events. The KEYNOTE-867 trial [30,31], which assessed concurrent pembrolizumab with SBRT in early-stage NSCLC, was terminated early due to lack of efficacy and increased toxicity. Together, results of these studies call into question the hypothesized ICI–radiation synergy and highlight the need to further investigate and optimize timing and disease stage. Data from clinical trials that combine radiation and ICI for treating metastatic NSCLC remain scarce and limited. A phase 1 trial (n = 37) showed that treatment with SBRT-ICI administered simultaneously, compared to sequentially (used within one week of each other), upregulated key immune pathways and enhanced local and distant tumor responses, although survival differences were not significant [32]. Notably, 9/37 patients with treated or asymptomatic BMs survived through the study period, which was a striking contrast to the non-BM cohort (n = 28) in which approximately two-thirds died by the follow-up (Bestvina et al., 2022, Figure A1 [32]). This suggests that patients with NSCLC and controlled BM or small asymptomatic central nervous system lesions may not have worse survival outcomes. However, the study was underpowered, and it was unclear whether the patients with BM had received prior radiation. To date, no published trials have assessed CR combined with ICIs specifically for NSCLC patients with BM.

Currently, reports of efficacy and tolerability of combined ICI-CR treatment for BM primarily stem from retrospective series that focused on melanoma patients [33,34,35,36]. Among NSCLC-specific studies or studies that included NSCLC in the mixture, only a few patients received both CR and ICI (n < 90) [37,38,39,40,41,42,43,44,45,46] and usually were from a single institution [47,48] (Appendix A). Despite these limitations, the studies showed better locoregional disease control with concurrent use of CR and ICI [42] than with SRS [45,47] or WBRT [46] administered alone or with ICI monotherapy [49]; this suggests synergistic effects when treating BM. However, a major difficulty with interpreting these findings is that the definitions of “concurrent use” varied widely—e.g., 2 weeks [47], 4 weeks [39], 30 days [46], 1 month [42], 3 months [45], 5 months [48]; therefore, optimal timing of the therapies is unclear. Moreover, existing reviews and meta-analyses synthesizing findings of these studies did not address the variability in optimal treatment timing [49,50,51]. Some preclinical studies show that administering ICI during the initial SRS treatment, when tumor burden is high, may reduce the effects of ICIs [24] and thus the potential synergistic effects. Conversely, others argue for a closer temporal proximity between ICI and SRS (2–4 weeks) to align with the half-life of ICIs (12–27 days) [52], which may enhance the synergy. This unresolved timing dilemma leaves clinicians without evidence-based guidance for effectively sequencing ICI and CR treatments.

To address these gaps, we used a large population-based dataset to compare survival outcomes across three clinically meaningful time windows for ICI administration relative to CR (2 weeks, 30 days, and 31–180 days) within 6 months after a diagnosis of NSCLC and SBM. Our study represents a significant advancement over prior research. It is the first direct evaluation of timing of the ICI-CR combination, assessing intervals that have previously been shown to be associated with better survival. Also, it establishes the largest cohort, to date, of patients with NSCLC and SBM who were treated with both modalities (all previous studies included N < 90). Finally, it focuses on two critically understudied populations—patients with untreated SBM, who typically are excluded from clinical trials and are older adults (≥65 years; mean: 73 years) who comprise >70% of NSCLC cases [53], yet frequently are underrepresented in clinical trials [54].

## 2. Materials and Methods

### 2.1. Data Source

This study used the Surveillance, Epidemiology, and End Result (SEER) cancer registry data by National Cancer Institute (NCI), which captures data from 34.6% of patients with cancer in the US and contains information on demographic, tumor, and treatment-related covariates at the diagnosis of the primary malignancy. The methodology, structure, and data collection methods were described elsewhere [55,56]. NCI biannually links SEER data to Medicare enrollment, fee-for-service claims, and Medicare Advantage encounter data (SEER-Medicare), which facilitates research on treatment patterns and outcomes [57]. The SEER-Medicare linked data used in this study included SEER data from January 2010 to December 2017 and Medicare enrollment and fee-for-service claims data from January 2009 to December 2019.

### 2.2. Study Sample

The study cohort included patients who received a diagnosis of NSCLC and SBM between 2010 and 2017 and who were treated with both ICIs and CR (SRS and non-SRS) within 6 months of diagnosis, a window that captures initial treatments and minimizes the possibility of treatment for disease progression. Patients with NSCLC and SBM were identified by ICD-O-3 codes for primary malignant neoplasm of bronchia and lung with NSCLC histology [58] and with a binary indicator variable for the presence of BM at diagnosis, which was available in SEER from 2010 onward.

For our analyses, we selected data from records of patients who met the following criteria: 1) NSCLC diagnosis at age ≥65 years, (2) NSCLC identified as sole primary cancer, (3) evidence of BM at NSCLC diagnosis, (4) uninterrupted enrollment in Medicare Part A (inpatient) and Part B (outpatient and physician services) but not Medicare Part C (Medicare Advantage) for a at least minimum of 6 months before diagnosis (for assessment of baseline comorbidities and performance status) and continuing until “*follow-up start date”* (to ensure observation of complete treatment). Patient records were excluded on the basis of the following criteria: (1) diagnosis of small cell lung cancer or other histology of lung cancer, (2) diagnoses of multiple primary cancers during the study period, (3) diagnosed at autopsy or died within 30 days of diagnosis, (4) received any oral targeted therapies during the study period (afatinib dimaleate, alectinib HCl, ceritinib, crizotinib, brigatinib, erlotinib HCl, ceritinib, crizotinib, dabrafenib mesylate, erlotinib HCl, gefitinib, osimertinib mesylate, trametinib dimethyl) to ensure patient cohort with similar tumor characteristics, (5) not treated with ICI throughout the study period, and (6) did not receive both ICI and CR within 6 months after diagnosis. Treatment with ICIs approved for lung cancer by the U.S. Food and Drug Administration was documented through Medicare claim records by using the corresponding Healthcare Common Procedural Coding System (HCPCS) codes [58]. The specific ICI agents included pembrolizumab, nivolumab, ipilimumab, avelumab, atezolizumab, and durvalumab. CR (SRS or non-SRS) was identified by linking Common Procedural Terminology/HCPCS [58] codes for radiation procedures with diagnosis codes for secondary malignant neoplasm of brain (ICD-9 198.3x or ICD-10 C79.31) within the same claim, thereby confirming radiation targeted to brain. Patients receiving dual PD-1 and CTLA-4 blockades were excluded from analysis due to potential survival outcome differences compared to ICI monotherapy or ICI-chemotherapy regimens [59] and later approval (26 May 2020) of ICI combination for lung cancer [60] relative to our study period. Because SEER provides only the month and year of the primary cancer diagnosis, we established the precise date of diagnosis by identifying the first diagnosis from Medicare claims and matching it with the SEER month and year of diagnosis. Further information is provided in the Appendix B section.

### 2.3. Treatment Groups

We analyzed data from patients who received ICI and CR within 6 months after receiving a diagnosis of NSCLC and SBM. Focusing on ICI and CR, the first treatment with either ICI or CR after diagnosis was designated as the index treatment. If a patient switched to the other treatment (e.g., from ICI to CR or vice versa), then the new treatment was classified as subsequent treatment. The switch was marked by the last claim for index treatment and the first claim for subsequent treatment. The time interval between these two dates was calculated in days. Based on this interval, patients were classified into three study groups: ≤15 days, 16–30 days, and >30 days. Patients were followed from the start of the subsequent treatment (i.e., *follow-up start date*). For example, if a patient initially received CR and then received ICI, then the date of ICI initiation was designated as the follow-up start date. An illustrative example is provided in Figure 1.

### 2.4. Survival Outcome

The primary study outcome was OS time, defined as number of days from start of follow-up until death. For patients who remained alive, survival times were censored at first occurrence of (1) loss of enrollment in Medicare Parts A and B, (2) enrollment in Medicare Advantage (Part C), or (3) end of data availability (31 December 2019).

### 2.5. Covariates

Covariates (available in SEER-Medicare database) were selected for their relevance to survival and/or receipt of treatment, according to our review of the published literature [61,62,63] and based on recommendations from our clinical collaborators. Patient characteristics at diagnosis included age, sex (male/female), race/ethnicity (non-Hispanic White/other), marital status (married/not married), census tract level poverty of the patient’s residence (0–<5%, 5–<10%, 10–<20%, 20–100%, or unknown), and rurality of the patient’s county of residence (rural/urban). Variables related to primary tumors included other metastasis (bone, liver, or lung) and histology of NSCLC (adenocarcinoma, squamous cell carcinoma, other). Variables related to cancer treatment included type of CR (SRS or non-SRS), line of ICI treatment (first-line, ≥second-line), and receipt of chemotherapy and neurosurgical resection within the first 6 months after diagnosis. To account for differences in patients’ baseline health status, we used Medicare claims data to calculate two key measures: (1) Modified Charlson Comorbidity Index (excluding cancer) [64] and (2) a performance status (PS) scale that uses a validated claims-based algorithm, which is a proxy of the Eastern Cooperative Oncology Group (ECOG) [65,66]. These measures were calculated with data from the 6 months preceding diagnosis. Detailed methodology for calculation of the performance status scale is available in the Appendix B section.

### 2.6. Statistical Analysis

The patient characteristics of the three study groups were compared with standard t-tests for continuous variables and Chi-square tests for categorical variables whenever appropriate. Outcomes were evaluated with survival analysis techniques. Survival curves were calculated using the Kaplan–Meier (KM) method and between-group comparisons of curves were performed using a log rank test. Adjusted multivariable Cox proportional hazards (CPH) regression was used to determine hazard ratios while controlling for potential confounders.

We found that the KM curves overlapped twice with delayed separation, indicating a violation of the proportional hazards (PH) assumption. To account for this violation, we conducted a sensitivity analysis by classifying our sample as ICI-CR administered within 15 days and >15 days. We then estimated a multivariable CPH model with two “change points” [67] at 90 days and 300 days, which is around the time point where the KM curves crossed each other, to estimate separate hazard ratios before and after the crossing points. The “change points” were chosen based on AIC and BIC by comparing models with various change points around the time when the curves crossed; *p*-values for multiple comparisons were adjusted using Bonferroni correction.

Since previous studies focused only on SRS types of CR, we conducted a subgroup analysis of data from patients who received ICI with SRS by using an adjusted multivariable CPH model. Given the distinct clinical profiles of patients who received SRS vs. non-SRS radiation (e.g., recipients of whole-brain radiotherapy typically present with greater metastatic burden), we performed an exploratory subgroup analysis of non-SRS population by using similarly adjusted models. A *p* < 0.05 was considered statistically significant. All analyses were performed using SAS statistical software (version 9.4; SAS Institute Inc., Cary, NC, USA).

### 2.7. Validation Study

To externally validate our findings, we also performed additional analyses using TriNetX LIVE™, a web-based platform which provides access to de-identified electronic health records from healthcare organizations worldwide. This platform allows for real-time cohort building and preliminary outcome analysis across diverse patient populations. We detailed the methods and results for the TriNetX analysis are available in the Appendix A.

## 3. Results

After we applied the inclusion and exclusion criteria, a final study sample of 236 patients remained for the primary analysis (subsequent ICI/CR treatment received ≤15 days: n = 117; 16–30 days: n = 42; >30 days: n = 77). Figure 2 displays the sample selection flow chart. The median age (range) of the sample was 72 years (range: 65–90 years). The median time between ICI and CR was 16 days (range: 0–160 days) for the entire sample. Patients in the >30 days group were significantly more likely (80.52%; *p* < 0.001) to receive chemotherapy within 6 months of diagnosis compared to the ≤15 days group (26.5%) or the 16–30 days group (30.95%). The study groups were found to be similar with respect to most other covariates. Median follow-up time was 145 days (range: 3–1561) for the entire study sample. The time from diagnosis to index date (mean [SD]: 68.65 [35.17] vs. 90.76 [30.59] vs. 123.7 [31.79]; *p* ≤ 0.0001) was significantly shorter for patients in the ≤15 days group, compared to the other study groups (Table 1).

Median survival was shortest for the 16–30 days group (92 days; 95% CI: 54–291), followed by the ≤15 days group (134 days; 95% CI: 101–181), and was longest for the >30 days group (209 days; 95% CI: 130–306); however, the log rank test showed no significant differences (Figure 3). After adjusting for covariates, the hazard of death between the study groups was not statistically significantly different (Table 2). 

Under the “change point” CPH model, the hazard of death for the ≤15 days group was significantly lower than the >15 days group only after 300 days from the follow-up start date (Appendix A).

Similar to results of the primary analysis, hazard of death between the study groups was not significantly different after using KM curves, log rank test (Appendix A), and multivariable CPH models (Appendix A) for subgroup analyses of ICI and SRS cohorts and of ICI and non-SRS cohorts.

Findings from the TriNetX validation cohort were consistent with those from the SEER-Medicare, showing no significant overall survival differences across ICI–CR timing intervals using KM curves and log rank test (Appendix A).

## 4. Discussion

In this population-based retrospective study, we compared survival outcomes of US patients with NSCLC and SBM, ages 65 years or older, after receiving ICIs and CR 15 days, 16–30 days, and >30 days apart, with both treatments administered within 6 months after diagnosis. Our analyses showed no significant differences in OS after ICI and CR were administered at these different time intervals. Subgroups analyses of patients who received either SRS or non-SRS with ICI produced similar results. However, for the group that received ICI within 15 days of CR, a survival benefit was observed 300 days after the start of the follow-up. Importantly, we validated our findings using the TriNetX LIVE™ Platform, which provided a complementary real-world cohort. The consistency of results across two independent datasets strengthens the robustness and generalizability of our conclusions.

Our findings on OS in patients with NSCLC and BM who were treated with SRS and ICI align with results of one previous study but contrast with those of other studies, underscoring the impact of the variable definitions for concurrent therapy. Consistent with our results, Ahmed et al. (2017) [37] examined 17 patients with NSCLC and BM who received SRS within 6 months of ICI, and the adjusted multivariate analysis revealed no significant difference in OS. Additionally, their median interval between ICI and SRS for non-concurrent use (median: 1.6 months; range: 0.2–4.7) was comparable to that of our study’s groups with treatment intervals greater than 15 days (median: 1.4 months; range: 0–5.6), reinforcing that broader timing windows may not confer an advantage for OS. In contrast, other studies suggest that survival improves with stricter concurrent timing. Schapira et al. (2018) [42] examined 37 patients with NSCLC and BM who were treated at a single cancer institution, and their results showed that 1-year OS increased when SRS was delivered within 1 month of ICI; however, Chen et al. [47] linked improved OS to a 2-week concurrent treatment window, compared to sequential treatment. Unlike our study, these two studies included patients with mixed histology, and the sample size for NSCLC was limited.

This lack of consensus likely stems from discrepancies in optimal timing and study methodology. The aforementioned studies included less uniform populations—patients who had NSCLC together with SBM (BM at diagnosis) or metachronous BM (developed after diagnosis); our study included only patients who had NSCLC with SBM, a group that is reported to have shorter survival than those with NSCLC and metachronous BM [68]. Moreover, our population includes exclusively patients who were at least 65 years of age at diagnosis. Thus, patients in our study may have succumbed to their systemic disease or to competing risk factors for death, which may partially explain why our results did not show OS benefits with ICI and CR across intervals. Additionally, chemotherapy exposure was higher in our >30-day group (62% vs. 48% for ≤15 days), which studies suggest could change the tumor immune microenvironment [69]. However, when we adjusted for chemotherapy use in our multivariable analysis, the results were not affected.

While no statistically significant differences were found across the intervals, a delayed survival benefit, approximately 300 days after the combination treatment, emerged among patients who received both treatments within 15 days of each other, which warrants further investigation. Preclinical evidence suggests that this delayed effect may stem from the time required for radiation-induced cell death to activate systemic T-cell responses [24,52], as well as the overlap between the pharmacokinetics of ICIs, which have a half-life of 12–27 days [52], and the window of radiation-induced PD-L1 upregulation, which peaks 3–7 days after radiation therapy and lasts up to 30 days [23,24]. However, caution should be exercised when interpreting the effect due to the smaller sample sizes after 300 days (Figure 3) and so larger studies are needed to investigate the validity of this finding.

Our study has several strengths. First, it addresses the underrepresentation of real-world NSCLC populations in the literature; most previous retrospective studies were limited to single-center cohorts. Secondly, existing evidence on the combination of ICI and CR primarily focuses on SRS, even though focal radiation, compared to WBRT, may increase the risk of distant brain failure in patients with NSCLC [70]. Our study included patients who received both SRS and non-SRS CR, which better captures real-world treatment patterns for NSCLC and SBM populations. Lastly, we evaluated the synergy of ICI and CR across three different time intervals, offering valuable insights into the impact of varied treatment timing on patient outcomes.

However, some limitations must be considered. First, although we examined various time windows for ICI and CR combination treatment, there may still be uncertainty about the effects of radiation on T-cell activation. Second, important prognostic factors such as smoking, EGFR/ALK mutations, and performance status were not available in our data. Nonetheless, we excluded patients who received NSCLC-specific targeted therapies to mitigate potential mutation-related confounding, and we used a validated proxy measure for performance status [65,66]. Third, limited sample size prevented separate analyses of the sequencing of CR before and after ICI and of subgroups with comorbidities that may influence treatment choices and outcomes. Accurate sequencing may be an inherent limitation because patients may receive ICI after CR, likely due to systemic progression, or receive CR after ICI, likely due to intracranial progression. Also, the effectiveness of treatment can be impacted by comorbid conditions, which are common in this elderly population (mean age 73) with NSCLC. For instance, COPD may limit the use of aggressive local therapy due to pneumonitis risk [71], and cardiovascular disease could delay ICI initiation due to concerns of myocarditis [72]. Further, non-cancer deaths from conditions like diabetes or renal disease may obscure true cancer-specific survival benefits, especially over longer follow-up. While we have adjusted for the overall comorbidity burden by using the modified Charlson Comorbidity Index, which includes these conditions, we could not separately assess whether optimal timing may differ according to these important comorbid conditions. Fourth, to define our cohort of SBM patients and indicators for other metastases at diagnosis, we used the SEER variable for metastasis sites at diagnosis; however, these variables have not been fully vetted [73]. Lastly, the inherent observational framework of this study may introduce immortal time bias, where patients with longer gaps between ICI and CR may appear to live longer simply because they survived long enough to receive both treatments. To reduce this risk, we required all patients to have received both treatments within 6 months of diagnosis, and their survival was followed from the start of the second treatment. Nonetheless, residual bias may remain if patients with longer time between receiving the first and the second treatment (e.g., 31–180 days) differ clinically from those who received them closer together (≤15 days), despite adjustment for key demographic and clinical prognostic factors. For instance, confounding by indication remains a concern because gaps between ICI and CR may be shorter for younger, healthier patients, whereas older, sicker patients might experience treatment delays due to recovery needs. Still, restrictions on patients receiving both treatments within 6 months, along with adjustment for age, comorbidities, and a proxy frailty score, should mitigate these biases and limit their impact on our comparative findings.

## 5. Conclusions

This population-based study of older patients with NSCLC and SBM showed no significant difference in OS when ICI and CR were administered within ≤15 days, 16–30 days, or 31–180 days after diagnosis. Notably, we observed a potential delayed survival benefit 300 days after the combination therapy among patients who received ICI ≤15 days of CR, suggesting that early combination therapy may be advantageous. However, the number of patients remaining in each group after 300 days resulted in small samples for this analysis, so large prospective studies are needed to validate this finding and to investigate potential heterogeneity among patients with comorbidities or with newer immunotherapy regimens.

## Figures and Tables

**Figure 1 curroncol-32-00499-f001:**
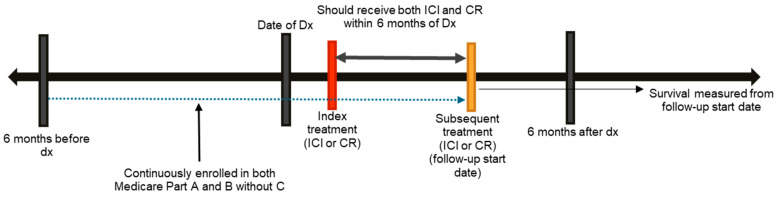
Illustrative example of study design. Abbreviations: Dx: diagnosis; ICI: immune checkpoint inhibitors; CR: cranial radiation.

**Figure 2 curroncol-32-00499-f002:**
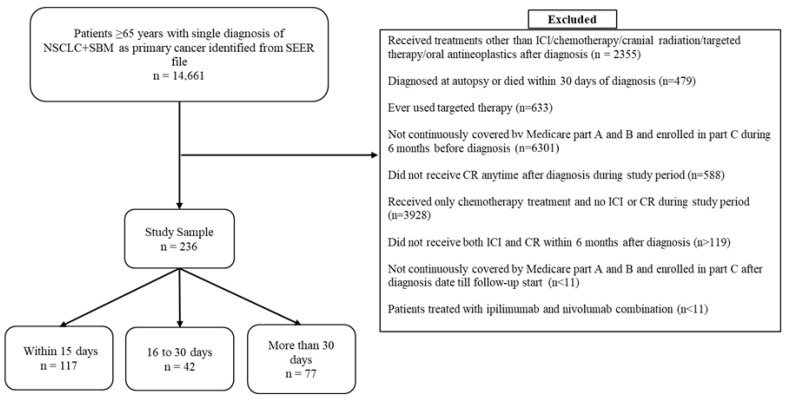
Sample selection flow chart. Abbreviations: SEER: Surveillance Epidemiology and End Results, SBM: synchronous brain metastases, NSCLC: non-small cell lung cancer, ICI: immune checkpoint inhibitors, CR: cranial radiation.

**Figure 3 curroncol-32-00499-f003:**
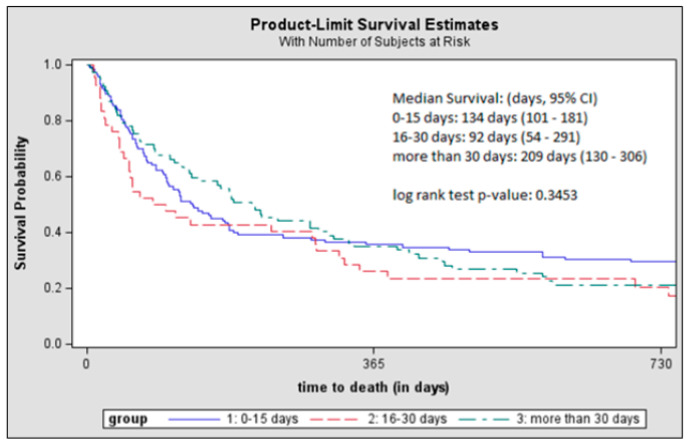
Kaplan–MeierKaplan-Meier survival curves of patients who received ICI and CR ≤ 15 days, 16 to 30 days, and >30 days apart. Kaplan–MeierKaplan-Meier curves are truncated at 730 days due to small sample sizes (all groups N < 11 after this point). Numbers at risk are reported where N ≥ 11; smaller values are suppressed as “<11” per SEER-Medicare data-use agreements. *p*-values are significant at α ≤ 0.05. Abbreviations: ICI, immune checkpoint inhibitors; CR, cranial radiation.

**Table 1 curroncol-32-00499-t001:** Patient demographic and clinical characteristics.

	Subsequent ICI-CR Treatment Received						
	Within 15 Days		16 to 30 Days		More than 30 Days		
	n = 117		n = 42		n = 77		
	**Mean (SD)**		**Mean (SD)**		**Mean (SD)**		***p*-value**
Age at diagnosis (years)	73.6 (5.8)		73.3 (5.7)		72.1 (5.7)		0.2914
Time from diagnosis to first ICI (days)	64.6 (33.1)		84.9 (29.4)		123.7 (31.7)		<0.0001
Time from diagnosis to first radiation (days)	52.8 (32.5)		64.3 (35.0)		41.0 (21.3)		0.0100
Time from diagnosis to start of follow-up (days)	68.6 (35.1)		90.7 (30.5)		123.7 (31.7)		<0.0001
	**N**	**%**	**N**	**%**	**N**	**%**	
** Sex **							
Male	60	51.3	17	40.5	33	42.9	0.3505
Female	57	48.7	25	59.5	44	57.1	
** Race/Ethnicity **							
Non-Hispanic White	89	76.1	>31	_a	>66	_a	0.0944
Other	28	23.9	<11	_a	<11	_a	
** Marital status at diagnosis **							
Non-married	46	39.3	20	47.6	28	36.4	0.4814
Married	71	60.7	22	52.4	49	63.6	
** Census Tract Poverty Indicator **							
0–<5% poverty	26	22.2	>11	_a	>20	_a	0.4676
5–<10%	33	28.2	<11	_a	>19	_a	
10–<20%	33	28.2	<11	_a	>20	_a	
20–100% or unknown	25	21.4	<11	_a	<11	_a	
** Rurality of patient’s county of residence **							
Metropolitan	102	87.2	>31	_a	>66	_a	0.9196
Non-metropolitan	15	12.8	<11	_a	<11	_a	
** Lung metastases at diagnosis **							
No	93	79.5	>31	_a	<66	_a	0.6670
Yes	24	20.5	<11	_a	>11	_a	
** Bone metastases at diagnosis **							
No	79	67.5	26	61.9	55	71.4	0.5664
Yes	38	32.5	16	38.1	22	28.6	
** Liver metastases at diagnosis **							
No	98	83.8	>31	_a	<66	_a	0.7481
Yes	19	16.2	<11	_a	>11	_a	
** Baseline Charlson Comorbidity **							
0	59	50.4	<18	_a	<48	_a	0.1690
1	31	26.5	>14	_a	>11	_a	
≥2	27	23.1	<11	_a	<18	_a	
** Baseline ECOG performance status proxy **							
ECOG 0–2	>106	_a	>31	_a	>66	_a	0.2423
ECOG 3–4	<11	_a	<11	_a	<11	_a	
** First treatment after diagnosis **							
ICI	28	23.9	<11	_a	<11	_a	<0.0001
Radiation	89	76.1	>31	_a	>66	_a	
** SRS within 6 months after diagnosis **							
No	54	46.1	25	59.5	41	53.3	0.2903
Yes	63	53.9	17	40.5	36	46.7	
** Neurosurgical resection within 6 months after diagnosis **							
No	94	80.3	>31	_a	<66	_a	0.7259
Yes	23	19.7	<11	_a	>11	_a	
** Type of treatment **							
First-line ICI	105	89.7	>31	_a	<66	_a	<0.0001
Second or greater line ICI	12	10.3	<11	_a	>11	_a	
** Chemotherapy within 6 months after diagnosis **							
No	86	73.5	29	69.1	15	19.5	<0.0001
Yes	31	26.5	13	30.9	62	80.5	
** Year of diagnosis **							
2015	<11	_a	<11	_a	>11	_a	NA
2016	>25	_a	<18	_a	>18	_a	
2017	>18	_a	>14	_a	<48	_a	
** NSCLC histology **							
Adenocarcinoma	83	70.9	>14	_a	<48	_a	0.0528
Squamous cell carcinoma	16	13.7	<11	_a	>11	_a	
Other	18	15.4	<18	_a	<18	_a	
** Primary tumor grade **							
Grade I/Grade II (well differentiated/moderately differentiated)	<11	_a	<11	_a	<11	_a	0.7529
Grade III/Grade IV (poorly differentiated/undifferentiated; anaplastic)	>25	_a	<18	_a	>18	_a	
cell type not determined	>18	_a	>14	_a	<48	_a	
** Primary tumor laterality **							
Bilateral involvement/midline/one side/lateral origin unknown	<11	_a	<11	_a	<11	_a	0.2750
Right: origin of primary	>25	_a	<18	_a	>18	_a	
Left: origin of primary	>18	_a	>14	_a	<48	_a	

Abbreviations: ECOG, Eastern Cooperative Oncology Group; ICI, immune checkpoint inhibitor; CR, cranial radiation; NSCLC, non-small cell lung cancer; SRS: stereotactic radiosurgery. All *p*-values significant at α ≤ 0.05. _a: Per the SEER-Medicare data-use agreement, counts and percentages are masked/concealed due to a case count of <11.

**Table 2 curroncol-32-00499-t002:** Adjusted hazard ratios of overall survival from multivariate cox proportional hazard regression analysis.

Variables	HR (95% CI)	*p*-Value
**Timing of ICI-CR treatment**		
≤15 vs. >30 days	1.09 (0.71–1.66)	0.6724
16–30 vs. >30 days	1.51 (0.90–2.53)	0.1145
≤15 vs. 16–30 days	0.72 (0.45–1.14)	0.1629
**Age at diagnosis (years)**	1.03 (1.00–1.06)	0.0128
**Marital status at diagnosis**		
Not married	1	-
Married	1.07 (0.77–1.49)	0.6510
**Sex**		
Female	1	-
Male	0.73 (0.53–1.01)	0.0644
**Race/Ethnicity**		
Non-Hispanic White	1	-
Other	0.81 (0.50–1.32)	0.4153
**Census tract poverty indicator**		
0–<5%	1	-
5–<10	1.08 (0.69–1.68)	0.7156
10–<20	0.82 (0.53–1.28)	0.3981
20–100	1.63 (0.95–2.80)	0.0757
unknown	1.14 (0.55–2.34)	0.7146
**Rurality of patient’s county of residence**		
Metropolitan area	1	-
Non-metropolitan area	0.82 (0.50–1.34)	0.4333
**Metastases at diagnosis**		
Bone metastasis	1.45 (1.04–2.02)	0.0270
Liver metastasis	0.96 (0.64–1.44)	0.8616
Lung metastasis	1.43 (0.99–2.07)	0.0542
**Charlson comorbidity index score**		
0	1	-
1	0.89 (0.60–1.32)	0.5887
≥2	0.87 (0.58–1.32)	0.5361
**Baseline ECOG performance status proxy**		
0–2	1	-
3–4	0.83 (0.30–2.28)	0.7309
**Histology**		
Adenocarcinoma	1	-
Squamous cell	0.81 (0.53–1.24)	0.0092
Other type	1.81 (1.15–2.83)	0.3408
**Tumor grade**		
1–2	1	-
3–4	0.90 (0.49–1.63)	0.7323
undetermined	1.24 (0.70–2.21)	0.4516
**Treatments within 6 months from diagnosis**		
SRS	0.67 (0.49–0.93)	0.0170
Chemotherapy	1.32 (0.89–1.95)	0.1578
Neurosurgical resection	0.58 (0.37–0.91)	0.0180

Abbreviations: HR, hazard ratio; CI, confidence intervals; ICI, immune checkpoint inhibitors; CR, cranial radiation; ECOG, Eastern Cooperative Oncology Group; SRS, stereotactic radiosurgery. All *p*-values significant at α ≤ 0.05.

## Data Availability

This study used the linked SEER-Medicare database. The interpreta-tion and reporting of these data are the sole responsibility of the authors. The authors acknowledge the efforts of the National Cancer Institute; the Information Management Services (IMS), Inc.; and the Surveillance, Epidemiology, and End Results (SEERs) Program tumor registries in the creation of the SEER-Medicare database. The datasets used to conduct this study are available on approval of a research protocol from the National Cancer Institute. Instructions for obtaining these data are available at Process for Obtaining SEER-Medicare Data https://healthcaredelivery.cancer.gov/seermedicare/obtain/ (accessed on 28 August 2025).

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
