# Peer review of "Survival Outcomes of Immune Checkpoint Inhibitors in Conjunction with Cranial Radiation for Older Adults with Non-Small Cell Lung Cancer and Synchronous Brain Metastasis"

_curroncol, 2025, doi:10.3390/curroncol32090499_

Round 1

Reviewer 1 Report

Comments and Suggestions for Authors

In this manuscript, the authors present a valuable population-based analysis evaluating the timing between immune checkpoint inhibitor (ICI) and cranial radiation (CR) administration in elderly NSCLC patients with brain metastases. Importantly, the study addresses a clinically relevant question using a robust dataset (SEER-Medicare). The methodology and conclusions are generally well presented, and the findings provide a meaningful foundation for future research. However, several minor issues should be addressed to improve the clarity and overall quality of the manuscript.

  1. The introduction would benefit from further expansion to provide a broader clinical and biological context. In particular, key distinctions between non-small cell lung cancer (NSCLC) and small cell lung cancer (SCLC) should be discussed. Additionally, incorporating relevant and recent literature will help strengthen the rationale for this study. Suggested references include:

Multi-omic profiling highlights factors associated with resistance to immuno-chemotherapy in non-small-cell lung cancer.

A single-cell atlas reveals immune heterogeneity in anti-PD-1-treated non-small cell lung cancer.

Molecular Subtypes and Targeted Therapeutic Strategies in Small Cell Lung Cancer.

Immune Checkpoint Inhibitors +/- Chemotherapy for Patients With NSCLC and Brain Metastases: A Systematic Review and Network Meta-Analysis.

Emerging Targets in Non-Small Cell Lung Cancer

  1. Although the primary analysis did not show statistically significant differences in overall survival, the delayed survival benefit (~300 days) in patients receiving ICI within 15 days of CR is noteworthy. This observation should be more clearly highlighted and discussed in the conclusion.

  1. The definitions of “first CR (or ICI)” and “subsequent ICI (or CR)” are somewhat ambiguous. Clarifying this distinction earlier in the abstract would improve reader comprehension and avoid potential confusion.

  1. In lines 261 and 380, please revise the descriptions “p = <.0001” and “(<=15 days)” for clarity and correctness. The appropriate formatting should be “p ≤ 0.0001” and “(≤15 days)”.

Reviewer 2 Report

Comments and Suggestions for Authors

Overall, it is well written, but there are a few points that need to be revised. 

Overall comments:  

The study on the combination of immune checkpoint inhibitors (ICI) and radiotherapy in elderly patients with brain metastases from NSCLC is clinically intriguing. Additionally, the use of large-scale population-based data enhances its reliability. 
Overall, the paper is well-written, but it tends to be somewhat verbose, and there are several areas that could be improved. 

  1. Introduction.

The background information provided in the Introduction is somewhat lengthy and could be more concise to better highlight the study’s objectives and novelty. It might be beneficial to streamline the Introduction to more clearly emphasize the aims and novelty of this research.  

  1. Materials and Methods.

The use of SEER-Medicare as a data source is commendable; however, there are limitations to the data, including the lack of information on smoking, comorbidities, and the presence or absence of EGFR/ALK mutations. These are important prognostic factors that could potentially influence the results, so the absence of such data should at least be discussed. 

The use of the Charlson Comorbidity Index is considered appropriate; however, it would be preferable to include a discussion on how specific comorbidities may influence treatment choices and outcomes. 

  1. Results.

Although a subgroup analysis was conducted for patients who underwent SRS based on existing reports, it is expected that the status of brain metastases differs between SRS and non-SRS patients. Including a subgroup analysis for non-SRS patients would likely enhance understanding. 

  1. Discussion.

The paper is somewhat lengthy, with a substantial portion of the discussion devoted to comparisons with existing studies. 

Rather than that, it would be better to deepen the discussion on the lack of statistical significance, the mechanisms behind the observed survival benefit at 300 days in the group within 15 days, and the effects of the sequence of ICI and CR. 

It would be beneficial to consider the impact of chemotherapy on survival duration with regard to the higher proportion of patients receiving chemotherapy in the group exceeding 30 days. 

Regarding the dosing interval between ICI and CR, considering the duration of radiation’s immune-stimulating effects and the pharmacokinetics of ICI may help us better understand how the timing and sequence of treatment affect outcomes. 

  1. 5. Conclusions

In the discussion, it is suggested that potential benefits of early combination therapy are observed in long-term follow-up and that further validation through larger-scale studies is necessary. It might also be beneficial to include future directions in the conclusion. 

Tables 

The tables are quite difficult to read in their current form. It would be helpful to organize them more clearly and concisely to improve readability and understanding. 

Round 2

Reviewer 2 Report

Comments and Suggestions for Authors

We acknowledge the revisions made and appreciate your thorough response to our suggestions. Overall, the manuscript, figures, and tables have been improved. However, we believe that Supplementary Table 1 still requires further enhancement.

Author Response

Thank you for appreciating our edits and providing valuable feedback which helped significantly improvethe quality and robustness of our manuscript. In response to your request we have edited the Supplementary table 1 to improve redability. The revised table can be found in the supplementary files attached.